# Phytochemical Profiling of *Sambucus nigra* L. Flower and Leaf Extracts and Their Antimicrobial Potential against Almond Tree Pathogens

**DOI:** 10.3390/ijms24021154

**Published:** 2023-01-06

**Authors:** Eva Sánchez-Hernández, Joaquín Balduque-Gil, Vicente González-García, Juan J. Barriuso-Vargas, José Casanova-Gascón, Jesús Martín-Gil, Pablo Martín-Ramos

**Affiliations:** 1Department of Agricultural and Forestry Engineering, ETSIIAA, Universidad de Valladolid, 34004 Palencia, Spain; 2AgriFood Institute of Aragon (IA2), Universidad de Zaragoza–CITA, 50059 Zaragoza, Spain; 3Department of Agricultural, Forest and Environmental Systems, Agrifood Research and Technology Centre of Aragón, Instituto Agroalimentario de Aragón—IA2, CITA–Universidad de Zaragoza, 50059 Zaragoza, Spain; 4Instituto Universitario de Investigación en Ciencias Ambientales de Aragón, EPS, Universidad de Zaragoza, 22071 Huesca, Spain

**Keywords:** black elderberry, *Diaporthe amygdali*, integrated pest management, *Phytophthora megasperma*, *Prunus dulcis*, *Verticillium dahliae*

## Abstract

Despite extensive research on the chemical composition of elderberries and their numerous uses in pharmaceutical, beverage, and food production, there is still a lack of knowledge about *Sambucus nigra* leaves and flowers’ antimicrobial activity against plant pathogens. In this study, the phytoconstituents of their aqueous ammonia extracts were first characterized by infrared spectroscopy and gas chromatography–mass spectrometry. The major phytocompounds identified in the flower extract were octyl 2-methylpropanoate; 3,5-dihydroxy-6-methyl-2,3-dihydropyran-4-one; propyl malonic acid; adenine; and 1-methyl-2-piperidinemethanol. Concerning the leaf extract, 1,6-anhydro-β-D-glucopyranose; oleic acid; 2,1,3-benzothiadiazole; 2,3-dihydro-benzofuran; and 4-((1E)-3-hydroxy-1-propenyl)-2-methoxyphenol and other phenol derivatives were the main constituents. The potential of the extracts to act as bioprotectants was then investigated against three almond tree pathogens: *Diaporthe amygdali*, *Phytophthora megasperma*, and *Verticillium dahliae*. In vitro tests showed higher activity of the flower extract, with EC_90_ values in the 241–984 μg·mL^−1^ range (depending on the pathogen) vs. 354–1322 μg·mL^−1^ for the leaf extract. In addition, the flower extract led to full protection against *P. megasperma* at a dose of 1875 μg·mL^−1^ in ex situ tests on artificially-infected excised almond stems. These inhibitory concentrations were lower than those of commercial fungicides. These findings suggest that *S. nigra* aerial organs may be susceptible to valorization as an alternative to synthetic fungicides for the protection of this important crop.

## 1. Introduction

*Sambucus nigra* L. (black elderberry or common elderberry) is a species of the *Adoxaceae* family (syn. *Caprifoliaceae*). It is a medicinal shrub or small tree native to the western and southern parts of Europe and North Africa. The species has also been introduced into other parts of the world, including North America, South East Asia, and Australia [1].

It has pinnate leaves, with five leaflets, short-stalked, elliptic, 4–12 cm long (Figure 1a). Inflorescences (Figure 1b) are small, yellowish-white, with three small bracts, and may have a peduncle; calyx minute, five-lobed; corolla five-cleft gamopetalous; five yellow stamens; and a tricarpellate pistil of three stigmas. Flowers have a strong pleasant odor, although they may have a fetid note [2], and are traditionally used to flavor wine, make tea and a nonalcoholic cordial, and are added to the batter used to prepare pastry [3]. Widely used in herbal medicine and the food industry, the fruits (Figure 1c) are shiny metallic, purple-black, 6–8 mm in diameter drupes that grow in corymbs containing several hundred pieces each [4].

The health benefits of the different parts of the *S. nigra* plant have been investigated by different authors. It has been found that elder flowers are antipyretic, diaphoretic, emetic, expectorant, anti-inflammatory, immunostimulant, antiviral, and antibacterial, while the leaves are disinfectant, diuretic, laxative, purify the blood, and have detoxifying properties [5,6].

The chemical components of *S. nigra* and its biological properties are tightly connected. The majority of the constituents of *S. nigra* flower aroma compounds are of terpenoid origin and include monoterpenes (phelandrene, *α-* and *γ*-terpinene, terpinolene, and safranal), terpenoid alcohols and oxides (terpinen-4-ol, hotrienol, α-terpineol, linalool, hydroxylinalool linalool oxides, *cis-* and *trans-*rose oxide, and nerol oxide), and a sesquiterpene (*β*-caryophyllene). Other terpenes—likely derivatives of carotenoids—are 6-methyl-5-hepten-2-one, 6-methyl-5-hepten-2-ol, *β*-ionone, *β*-damascenone, and 1,1,6-trimethyl−1,2-dihydronaphthalene (1,1,6-TDN). Another major group of the constituents arises from autoxidation of fatty acids, being primarily constituted of butanol, 2-hexenol, 3-hexenol, 3-hydroxy-2-butanone, 1-penten-3-one, 1-octen-3-one, 2,4-heptadienal, and 2-octenal [7]. Both flowers and leaves of black elder are a rich source of basic substances such as rutin, quercetin, protocateuchic acid, 3,5-dicaffeoylquinic acid, neochlorogenic acid, and tannins, as well as vitamin E [3]. On the other hand, sambunigrin and prunasin are the cyanogenic glycosides found in *S. nigra* that are present in the greatest amounts in all of its sections. Specifically, *S. nigra* leaves contain the highest concentrations of sambunigrin (27.68–209.61 µg·g^−1^ fresh weight), followed by flowers (1.23–18.88 µg·g^−1^ FW) and berries (0.08–0.77 µg·g^−1^ FW) [8]. *Sambucus nigra* also has m-hydroxysubstituted glycosides, such as zierin and holocalin. All these substances have the potential to be poisonous and lethal since they produce cyanide after hydrolysis [9].

Concerning the antimicrobial activity of *S. nigra*, and on the basis of the fact that all organs of the black elder contain the aforementioned cyanogenic glycosides prunasin and sambunigrin, it has been suggested that their plant extracts may possess antifungal activity [10]. Álvarez et al. [11] extracted cysteine-rich peptides from the flowers in a mixture of dicloromethane/methanol, which exhibited activity against different fish pathogens of aquaculture interest (viz., *Escherichia coli* (Migula) Castellani and Chalmers, *Vibrio anguillarum* Bergeman, *Vibrio ordalii* Schiewe et al., *Flavobacterium psychrophilum* (Bernardet and Grimont) Bernardet et al., and *Aeromonas salmonicida* (Lehmann and Neumann) Griffin et al.). It was postulated that their antimicrobial activity was related to bacterial cell membrane disruption. In 2013, a brief report regarding the antifungal activity of *S. nigra* fruit ethanolic extracts concluded the existence of in vitro inhibitory activity against *Phytophthora infestans* (Mont.) de Bary [12]. Binding of antifungal PR 1 proteins to the protein channels in cell membranes, affecting the release of Ca^2+^ ions, was proposed as the mechanism behind such antifungal activity [13].

As previously mentioned, there is still a lack of knowledge about the leaves’ and flowers’ bioactive compound content and potential to act as bioprotectants in fruticulture, specifically to control diseases associated with plants of the genus *Prunus* L. (cherry, almond, peach, and plum). In farmed *Prunus* plants, more than ten *Phytophthora* species have been found to cause root rot, crown rot, and stem and scaffold cankers. Among them, the soil-borne infection caused by *Phytophthora megasperma* Drechs. is frequently linked to root and crown rots, as well as trunk cankers [14], causing tree losses, especially of young plants in poorly drained soils.

Vascular wilts are among the most severe fungal diseases in the world, and certain species of the genus *Verticillium* are responsible for them. These soil-borne, cosmopolitan, ascomycetous fungi possess a wide range of plant hosts, causing significant yield losses [15]. *Verticillium dahliae* Kleb., one of the many species of the genus, can infect more than 200 plant species worldwide [16]. Verticillium wilt of almond (*Prunus dulcis* (Mill.) D.A.Webb) can damage developing orchards and reduce productivity. Although historical reports of Verticillium wilt of *Prunus* spp. usually describe individual branch or limb death, with rare losses of entire trees (usually in the second or third growing season), they result in substantial economic losses primarily due to tree removal and replacement cost, extra pruning, and lost production from weakened plant stands [17].

As for *Diaporthe* species—which can either be plant pathogens, endophytes, or saprobes—some have been linked to twig canker, bud and shoot blight, dieback, wood decay, and fruit rot in almond trees [18]; canker, shoot dieback, and bud and shoot blight in peaches [19]; cankers and shoot blight in apples [20]; and dieback and canker in pear and plum trees [21]. In particular, *Diaporthe amygdali* (Delacr.) Udayanga, Crous and K.D. Hyde has been identified as the causal agent of twig canker and blight in almond and peach trees, being associated with fruit rot of peaches and fruit rot and branch dieback (‘Fusicoccum canker’) of almond [18]. The rapid desiccation of buds, blooms, and leaves in late winter or early spring is one of the disease’s symptoms. Brown lesions, which first appear around green shoot buds, progress into yearly sunken cankers, which can occasionally have a sticky exudate as well as twig withering [22].

According to Article 14 of Directive 2009/128/EC, which promotes the use of formulations based on natural products in the frame of Integrated Pest Management programs, the current work suggests the use of aqueous ammonia extracts of the flowers or leaves of *S. nigra* as biorationals for the protection of certain important fungal pathogens of *P. dulcis*. Both extracts (whose main constituents were determined by GC–MS) were first studied in vitro against the aforementioned three pathogens, and the most active extract was subsequently tested in ex situ assays on excised stems to confirm its anti-oomycete activity against *P. megasperma*.

## 2. Results

### 2.1. Vibrational Characterization

The main infrared absorption bands present in the spectra of *S. nigra* flowers and leaves, as well as in those of their aqueous ammonia extracts (Appendix A), are summarized in Table 1.

The spectral profiles are compatible with the presence of the functional groups of alkaloids, polyphenols, and organic acid esters. For example, the key vibrations bands at 3369 cm^−1^ (indicative of O–H stretching of free hydroxyl groups), 2922 cm^−1^ (indicative of C–H stretching of alkanes), and 1602 cm^−1^ (indicative of *α* and *β* unsaturated ketone groups) are consistent with the functional groups found on DDMP (2,3-dihydro-3,5-dihydroxy-6-methyl-4H-pyran-4-one or 3,5-dihydroxy-6-methyl-2,3-dihydropyran-4-one), in agreement with the work by Olaniyan et al. [23].

### 2.2. Analysis of the Constituents of the Flower and Leaf Extracts by GC-MS

The results obtained from the phytochemical analyses of *S. nigra* flower extract (Appendix A) show relative high concentrations of octyl 2-methylpropanoate (also named octyl isobutyrate and caprylyl isobutyrate; 10.5%); DDMP (5.1%); 2-propyl malonic acid (or 2-propylpropanedioic acid) and dimethylmalonic acid, ethyl isohexyl ester (4.9%); heptanal-related compounds (4.6%); glycerin (4.2%); adenine (4.2%); and 1-methyl-2-piperidinemethanol (3.5%). Other minor constituents include 2-cyclopentylidene-cyclopentanone (3%); benzoic acid derivatives (2.5%); N,N-diamylmethylamine (2.3%); geranic acid (1.7%); catechol (1.6%); and 2,3-dihydro-benzofuran (1.6%).

In turn, the main phytochemicals identified from *S. nigra* leaf extract (Appendix A) were 1,6-anhydro-β-D-glucopyranose (11.6%); oleic acid (6.3%); butanoic acid pentyl ester or pentyl butyrate (8%); 2,1,3-benzothiadiazole (4.5%); trans-3-penten-2-ol (3.9%); 2,3-dihydro-benzofuran (3.5%); 4-((1E)-3-hydroxy-1-propenyl)-2-methoxyphenol and other phenol derivatives (3.1%); 6-hydroxy-4(1H)-pyrimidinone (2.9%); catechol (2.8%); inositol derivatives (2.2%); DDMP (2.1%); 2,3-dihydroxycyclohexanone (2.0%); glycerin (1.8%); and hydroquinone (1.7%).

Hence, the flower and leaf extracts shared the presence of glycerin, DDMP, 2,3-dihydro-benzofuran, catechol, and hydroquinone.

### 2.3. Antifungal and Anti-Oomycete Activity Assessment

#### 2.3.1. In Vitro Activity

The aqueous ammonia extracts of flowers and leaves of *S. nigra* were evaluated for their capacity to inhibit the mycelial development of *D. amygdali*, *P. megasperma*, and *V. dahliae* strains (Figure 2). The flower extract of *S. nigra* was more effective than the leaf extract against *D. amygdali*, with inhibition values of 1000 and 1500 µg·mL^−1^, respectively, and equally effective against the oomycete *P. megasperma* and the soil-borne ascomycete *V. dahliae*, inhibiting their mycelial growth at concentrations of 375 and 1500 µg·mL^−1^, respectively.

Because of the higher activity of the flower extract, and in a first approach to investigate its mode of action, three of its constituents (viz., DDMP, octyl isobutyrate, and 1-methyl-2-piperidine methanol, shown in Figure 3) were also tested. DDMP was equally effective against *P. megasperma* and *V. dahliae*, with inhibition values of 375 µg·mL^−1^, compared to 750 µg·mL^−1^ for *D. amygdali*. The most abundant compound in the flower extract, octyl isobutyrate, was most effective against *P. megasperma* (MIC = 250 µg·mL^−1^), followed by *V. dahliae* (MIC = 500 µg·mL^−1^) and *D. amygdali* (MIC = 750 µg·mL^−1^). Finally, 1-methyl-2-piperidinemethanol inhibition was higher in *P. megasperma* (MIC = 375 µg·mL^−1^) compared to that obtained against *V. dahliae* (MIC = 500 µg·mL^−1^) and *D. amygdali* (MIC = 750 µg·mL^−1^).

Table 2 summarizes the effective concentrations (EC_50_ and EC_90_) obtained for the two extracts and each of the pure compounds against each pathogen.

For comparison purposes, three commercial synthetic fungicides (viz., azoxystrobin, mancozeb, and fosetyl-Al) were also tested at three concentrations (the recommended dose, 1/10th of the recommended dose, and 10 times the recommended dose). Radial growth inhibition results are summarized in Table 3. The highest activity was attained by mancozeb, which fully inhibited the growth of *V. dahliae*, *D. amygdali*, and *P. megasperma* at doses of 150, 1500, and 15,000 µg·mL^−1^, respectively. Fosetyl-Al showed an intermediate activity, achieving full inhibition at 2000 µg·mL^−1^ against *V. dahliae*, and at 20,000 µg·mL^−1^ against the other two pathogens. The lowest activity was attained by azoxystrobin, which did not fully inhibit the growth of *P. megasperma* at ten times the recommended dose, and for which MIC values of 625,000 µg·mL^−1^ were obtained against *D. amygdali* and *V. dahliae*.

#### 2.3.2. Ex Situ Activity for the Protection of Excised Stems

Because of the high activity of *S. nigra* extracts, ex situ tests were conducted on almond rootstock ‘Garnem’ excised stems to evaluate the efficacy of the most active treatment (the flower extract) against the most resistant pathogen, viz., *P. megasperma*. At the lowest assayed concentration, i.e., the MIC value obtained in the in vitro tests (375 μg·mL^−1^), no protection was observed, with canker lengths similar to those of the untreated stems. When the dose was increased by a factor of 2.5 (937.5 μg·mL^−1^), large cankers were still registered, with no significant differences versus the control. Only when the protective treatment concentration was five times the MIC value (1875 μg·mL^−1^) was full protection of the excised stems achieved, with no signs of fungal colonization in the outer bark or in the cambium tissues of any of the replicates (Figure 4).

To fulfill Koch’s postulates, samples of the inoculated ‘Garnem’ stems showing cankers were taken apart and mounted on a microscope slide with 3% KOH as mounting media and morphologically inspected to confirm the identity of the microorganism responsible for the lesions. Such microscopical observations confirmed the presence of somatic and reproductive structures compatible with those of *P. megasperma* (Figure 5).

## 3. Discussion

### 3.1. Phytochemical Profile

As indicated above, previous works on *S. nigra* flower aqueous extracts mainly reported the presence of terpenes—including monoterpenes, terpenoids alcohols and oxides, sesquiterpenes, and derivatives of carotenoids—and compounds arising from the autoxidation of fatty acids [7], none of them present in the extract studied herein (except for terpinen-4-ol and heptanal). Other reports on aqueous, ethanolic, and methanolic flower and leaf extracts detected rutin, quercetin, protocateuchic acid, 3,5-dicaffeoylquinic acid, neochlorogenic acid, tannins, vitamin E, and glycosides [3,8,9], out of which only the latter were found in the leaf aqueous ammonia extract (e.g., 1,6-anhydro-β-D-glucopyranose) here analyzed. Although the composition of *S. nigra*’s components is known to be influenced by a number of variables, including variety, cultivar, and environmental and climatic circumstances [24], in our view, differences, in this case, should be mainly ascribed to the choice of the extraction solvent. Nonetheless, as discussed below, the presence of the main compounds identified herein has also been reported in other plant extracts.

Concerning the flower extract constituents, octyl isobutyrate, the octyl ester of isobutyric acid, was identified as one of the major phytochemicals in essential oils with antimicrobial potential from *Malabaila aurea* Boiss. aerial parts (40%) [25], and is also present in large amounts in the bark from *Uncaria tomentosa* (Willd. ex Schult.) DC. (30.7%) [26]; in *Heracleum sphondylium* subsp. *ternatum* (Velen.) Brummitt (24.6%) [27]; in *Heracleum persicum* Desf. (17.82%) [28]; and—in smaller proportions—in the essential oils from roots of *Caucalis platycarpos* L. (8.5%), *Elaeosticta glaucescens* Boiss. (4.0%), and *Eryngium caucasicum* Trautv. (2.8%) [29].

Due to their extensive biological and pharmaceutical properties, 4H-pyrans belong to a significant class of heterocyclic compounds. In particular, the flavonoid named 4H-pyran-4-one, 2,3-dihydro-3,5-dihydroxy-6-methyl- (or 3,5-dihydroxy-6-methyl-2,3-dihydropyran-4-one) has been isolated from *Chuanminshen violaceum* Sheh and Shan (25%) [30], *Punica granatum* var. *nana* L. (9.7%) [31], *Nephrolepis biserrata* (Sw.) Schott (9.3%) [32], *Hibiscus syriacus* L. (4.2%) [33], *Cyperus rotundus* L. (3.8%) [34], *Euphorbia serrata* L., *Camellia japonica* L., *Acalypha indica* L., *Ammannia baccifera* L., *Borassus flabellifer* L., *Cocculus hirsutus* (L.) Diels, *Cucumis sativus* L., *Leucas aspera* (Willd.) Link, *Litchi chinensis* Sonn., *Marsilea quadrifolia* L., and *Rumex vesicarius* L. [35]. It has been reported to display antioxidant, antimicrobial, anticancer, anti-inflammatory, and cytotoxic activities [31,36,37].

Although acid 2-propylmalonic acid (a fatty acid derivative) has not been reported to display antimicrobial activity, its diisopropyl malonate derivative is used for the synthesis of the fungicide isoprothiolane [38]. Due to their numerous applications, piperidine alkaloid compounds are among the chemical classes that are often studied in medicine as raw ingredients to create new medications or as medications themselves to treat certain ailments [39].

The presence of glycerin in plant extracts has been reported, for instance, in *Plantago major* L. leaf extracts [40]; in *Allamanda cathartica* L. [41]; in *Cynodon dactylon* (L.) Pers. (with an activity comparable to that of streptomycin against *Staphylococcus aureus* Rosenbach, *E. coli, Salmonella typhi* (Schroeter) Warren and Scott, *Proteus mirabilis* Hauser, and *Streptococcus pyogenes* Rosenbach) [42]; in *Salvadora persica* L. (with activity against *S. aureus* and *Aspergillus terreus* Thom) [43]; and in *Aphelandra squarrosa* Nees (with a glycerin content as high as 46% and strong activity against *E. coli*) [44].

As for 1-methyl-2-piperidinemethanol, it has been previously isolated from young stems of *Lobelia polyphylla* Hook. and Arn. (11.8%) [45], as well as in *Artemisia alba* Turra (2.1%) [46]. It is used as a reagent to synthesize phenylpyridone derivatives, compounds that act as anti-obesity agents in mice [47], but—to the best of our knowledge—no previous reports on its antimicrobial activity are available.

Apropos of the main constituents present in the leaf extracts, 1,6-anhydro-*β*-D-glucopyranose (levoglucosan) may be considered an artifact resulting from the degradation of cellulose during the extraction. Oleic acid, widely present in other leaf extracts, such as those from *Sesuvium portulacastrum* L. [48], has been reported to have antifungal activity against two plant pathogenic fungi, *Pythium ultimum* (Trow) Uzuhashi, Tojo and Kakishima and *Crinipellis perniciosa* (Stahel) Singer [49]. Regarding butanoic acid pentyl ester or amyl butyrate (related to octyl butyrate), a natural product found in strawberry, banana, apple, and apricot fruits, it has been qualified as a safe chemical [50], but no biological activity has been reported to date.

### 3.2. Antimicrobial Activity Comparison

#### 3.2.1. Comparison with Other *S. nigra* Extracts

Compared to elderberry fruits, flowers and leaves have not garnered as much attention for their antimicrobial properties. Appendix A shows previous research on the potential of *S. nigra* flower and leaf extracts.

Ferreira-Santos et al. [51] chemically characterized an aqueous extract of *S. nigra* flowers and tested its antimicrobial potential, highlighting the inhibition obtained against the Gram-positive bacteria *S. aureus* ATCC 25293 and *Staphylococcus epidermidis* (Winslow and Winslow) Evans ATCC 12228, at 8300 and 4100 µg·mL^−1^, respectively. Moderate activity was also obtained by Caroline et al. [52] against methicillin-resistant *S. aureus* (MRSA) using a hydroalcoholic extract of flowers, with an inhibition zone (IZ) of 17 mm, the highest obtained against the microorganisms tested. However, they noted that the aqueous leaf extract had no activity against MRSA. In the same line, Cioch et al. [53] indicated that ethanolic, methanolic, and/or aqueous extracts of *S. nigra* flowers minimally inhibited the 11 species of microorganisms (eight bacteria and three fungi) studied. 

Regarding phytopathogens, Schoss et al. [54] performed a supercritical fluid extraction of flowers using CO_2_, finding that the extract inhibited the growth of *Fusarium poae* (Peck) Wollenw. and *Botrytis cinerea* Pers. by 75 and 81%, respectively, at a 10% concentration. Such a dose was several orders of magnitude higher than those reported herein, but—given that the pathogens were different—a direct comparison is not possible.

#### 3.2.2. Comparison of Efficacy vs. Other Natural Compounds

If a comparison with other natural compounds in terms of activity against the same phytopathogens is attempted (Appendix A), no data are available—to the best of the authors’ knowledge—against *D. amygdali*. In the case of *P. megasperma*, an EC_90_ value of 422 µg·mL^−1^ was reported by Ramadan et al. [55] for an *Ageratum houstonianum* Mill. essential oil, lower than those of *S. nigra* flower and leaf extracts (982 and 1322 µg·mL^−1^, respectively). As for *V. dahliae*, many more plant-derived products have been tested in the literature. The MIC values reported herein for *S. nigra* extracts (984 and 975 µg·mL^−1^) would be approximately twice those of *U. tomentosa* [26] and *Chrysanthemum coronarium* L. [56] extracts; would be comparable to the one reported for *Haplophyllum tuberculatum* (Forssk.) Juss. extract [56]; and would be substantially better than those reported for extracts from *Origanum heracleoticum* Benth., *Salvia officinalis* L., *Rosmarinus officinalis* L., *Mentha piperita* L., *Thymus vulgaris* L., and *Lavandula angustifolia* Mill., among others [57,58,59].

#### 3.2.3. Comparison with Conventional Fungicides

If the values reported in Table 2 and Table 3 are compared, it may be observed that the in vitro fungicidal activities of *S. nigra* flower and leaf aqueous ammonia extracts were higher than those of the three assayed conventional synthetic fungicides against both *D. amygdali* and *P. megasperma*. MIC values of 1000 and 1500 µg·mL^−1^ were registered against *D. amygdali* for the flower and leaf extracts, respectively, vs. 1500, 2000, and 625,000 µg·mL^−1^ for mancozeb, fosetyl-Al, and azoxystrobin, respectively. As for *P. megasperma*, full inhibition was achieved at 375 µg·mL^−1^ for the two *S. nigra* extracts, whereas doses of 15,000 and 20,000 µg·mL^−1^ were needed in the case of the dithiocarbamate and organophosphorus fungicides, respectively (and the strobilurin fungicide did not achieve full inhibition even at ten times the recommended dose, 625,000 µg·mL^−1^). Concerning *V. dahliae*, for which full growth inhibition was attained at a concentration of 1500 µg·mL^−1^ in the case of the two *S. nigra* extracts, it is worth noting that a better activity was registered for mancozeb (with a ten times lower MIC value, of only 150 µg·mL^−1^), but the natural products still performed better than fosetyl-Al (MIC = 2000 µg·mL^−1^) and azoxystrobin (MIC = 625,000 µg·mL^−1^).

Should the results of the ex situ assays against *P. megasperma* be compared instead, the dose at which full protection was achieved (1875 μg·mL^−1^) would still be an order of magnitude lower than those required for mancozeb and fosetyl-Al. Consequently, *S. nigra* aerial part extracts may be regarded as particularly promising biorrationals against this pathogen.

#### 3.2.4. Mode of Action

Although an in-depth study of the underlying mechanism of action is beyond the scope of the work presented herein, the antimicrobial activity observed both in vitro and ex situ for *S. nigra* extracts should be mainly attributed to the three constituents that have been separately tested as pure reagents (viz., octyl isobutyrate, 4H-pyran-4-one, 2,3-dihydro-3,5-dihydroxy-6-methyl- (DDMP), and 1-methyl-2-piperidine methanol; Figure 3), although contributions from other phytoconstituents and/or synergies among constituents cannot be ruled out. This hypothesis is supported by numerous studies that have also detected these compounds, or derivatives thereof, in other plant extracts or essential oils, and that have also reported associated antimicrobial activities (summarized in Appendix A).

High contents of octyl isobutyrate, the octyl ester of isobutyric acid, have been identified in plants of the *Apiaceae* family. Hamedi et al. [29] characterized their content and evaluated the antimicrobial activity of essential oils from roots of three Iranian endemic plants—*Elaeosticta glaucescens* Boiss (4%), *Caucalis platycarpos* L. (8.5%), and *Eryngium caucasicum* Trautv. (2.8%)—against *E. coli*, *Pseudomonas aeruginosa* (Schroeter) Migula, *Staphylococcus aureus* Rosenbach, and *Bacillus subtilis* (Ehrenberg) Cohn, finding MIC values in the 500–1000 µg·mL^−1^ range. İşcan et al. [27] studied the essential oil of *Heracleum sphondylium* subsp. *ternatum* (Velen.) Brummitt (24.6%) and tested it against 17 human and plant pathogens. The MIC values against *E. coli*, *P. aeruginosa*, and *S. aureus* were in agreement with the above-mentioned range. Inhibition of *Listeria monocytogenes* (Murray et al.) Pirie (MIC = 125 µg·mL^−1^) was higher than that obtained for the essential oil of *Heracleum persicum* Desf. (17.82%) [28,60], but inhibition of *Candida albicans* (C.P. Robin) Berkhout (MIC = 500 µg·mL^−1^) was lower than that obtained by Tzakou et al. [25] using an essential oil of aerial parts of *Malabaila aurea* Boiss. (40%). Finally, and with respect to phytopathogenic bacteria, the essential oil of *H. sphondylium* subsp. *ternatum* inhibited the genus *Pseudomonas* with MICs in the 31.25–500 µg·mL^−1^ range, while *Xanthomonas* sp. was inhibited at 31.25 µg·mL^−1^.

4H-pyran derivatives are widely distributed in natural compounds [61]. They are characterized by numerous biological and pharmaceutical activities such as antifungal, antiviral, antioxidant, antileishmanial, antiallergic, antibacterial, hypotensive, anticoagulant, diuretic, and antitumor activities [62,63]. Regarding 4H-pyran-4-one, 2,3-dihydro-3,5-dihydroxy-6-methyl-, our group previously studied a hydromethanolic extract of *P. granatum* var. *nana* fruits, with a DDMP content of 9%, against woody crop phytopathogens, with inhibition values of 1500 µg·mL^−1^ [31]. Alghamdi et al. [64] used extracts of leaves and bark of *Eucalyptus camaldulensis* Dehnh., with a DDMP content of up to 3%, against two Gram-negative bacteria (*P. aeruginosa* and *E. coli*) and two Gram-positive bacteria (*B. subtilis* and *S. aureus*), with inhibition values in the 0.391–25 µg·mL^−1^ range.

In relation to piperidine-derived alkaloids, which are generally obtained from *Piper nigrum* L. and *Conium maculatum* L. [65], they have many pharmacological activities including anticancer, antibacterial, antidepressant, herbicide, antihistamine, central nervous system stimulant, insecticide, and fungicide action [66]. As for 1-methyl-2-piperidinemethanol, Riahi et al. [67] reported contents of 1-methyl-2-piperidinemethanol of 2.66–3% (lower than that obtained in *S. nigra* flowers) in *Mentha* × *rotundifolia* (L.) Huds. leaf essential oil, with a high bactericidal activity against Gram-negative bacteria *E. coli* and *Salmonella typhimurium* (Loeffler) Castellani and Chalmers. Other authors have tested artificial chemical derivatives of piperidine, although with worse inhibition values [68,69].

## 4. Materials and Methods

### 4.1. Reagents

Octyl-2-methylpropanoate, ≥99% (CAS No. 109-15-9); 3,5-dihydroxy-6-methyl-2,3-dihydropyran-4-one (CAS No. 28564-83-2); 1-methyl-2-piperidinemethanol, 98% (CAS No. 20845-34-5); acetic acid, glacial, ACS reagent, ≥99.7% (CAS No. 64-19-7); and ammonia, anhydrous, ≥99.98% (CAS 7664-41-7) were acquired from Sigma-Aldrich Química (Madrid, Spain). Potato dextrose agar (PDA) was purchased from Becton Dickinson (Bergen County, NJ, USA).

Commercial fungicides used for comparison purposes, viz., Ortiva^®^ (azoxystrobin 25%; reg. no. 22000; Syngenta), Vondozeb^®^ (mancozeb 75%; reg. no. 18632; UPL Iberia), and Fosbel^®^ (fosetyl-Al 80%, reg. no. 25502; Probelte) were kindly provided by the Plant Health and Certification Service (CSCV) of Gobierno de Aragón.

### 4.2. Plant Material and Extraction Procedure

*Sambucus nigra* samples were collected in June 2022, in Alerre (Huesca, Spain; 42°09′27.4″ N 0°27′50.6″ W). A voucher specimen, identified and authenticated by Prof. J. Ascaso, has been deposited at the herbarium of the Escuela Politécnica Superior, Universidad de Zaragoza (Huesca, Spain). Aerial parts from different specimens (*n* = 20) were thoroughly mixed to obtain (separate) flowers and leaves composite samples. The composite samples were shade-dried, pulverized to a fine powder in a mechanical grinder, homogenized, and sieved (1 mm mesh).

An aqueous ammonia solution was chosen to dissolve the bioactive compounds of interest. The flower extract was prepared according to the procedure described in [26]: the flowers powder (30 g) was first digested in an aqueous ammonia solution (140 mL H_2_O + 10 mL NH_3_) for 2 h, then sonicated in pulsed mode (with a 2 min stop every 2.5 min) for 10 min using a probe-type ultrasonicator (model UIP1000hdT; 1000 W, 20 kHz; Hielscher Ultrasonics, Teltow, Germany), and then allowed to stand for 24 h. It was then adjusted to neutral pH using acetic acid. Finally, the solution was centrifuged at 9000 rpm for 15 min, and the supernatant was filtered through Whatman No. 1 paper. The extraction procedure for leaf samples was identical. 

Aliquots of both extracts were freeze-dried for Fourier transform infrared (FTIR) analyses.

### 4.3. Phytopathogens Isolates

The fungal isolates of *D. amygdali* (isolate MYC-765), *P. megasperma* (isolate MYC-1488) and *V. dahliae* (isolate MYC-1134) were supplied as subcultures in PDA plates by the Mycology lab of the Center for Research and Agrifood Technology of Aragón (CITA).

### 4.4. Characterization Procedures

The infrared vibrational spectra of the fresh flower and leaf samples, as well as those of the corresponding freeze-dried extract samples, were registered using a Thermo Scientific (Waltham, MA, USA) Nicolet iS50 Fourier transform infrared spectrometer, equipped with an in-built diamond attenuated total reflection (ATR) system. The spectra were collected over the 400–4000 cm^−1^ range, with a 1 cm^−1^ spectral resolution, taking the interferograms that resulted from co-adding 64 scans. 

The aqueous ammonia extract was studied by gas chromatography–mass spectrometry (GC–MS) at the Research Support Services (STI) at Universidad de Alicante (Alicante, Spain) using a gas chromatograph model 7890A coupled to a quadrupole mass spectrometer model 5975C (both from Agilent Technologies). The chromatographic conditions were injection volume = 1 µL; injector temperature = 280 °C, in splitless mode; initial oven temperature = 60 °C, 2 min, followed by ramp of 10 °C/min up to a final temperature of 300 °C, 15 min. The chromatographic column used for the separation of the compounds was an Agilent Technologies HP-5MS UI of 30 m length, 0.250 mm diameter, and 0.25 µm film. The mass spectrometer conditions were temperature of the electron impact source of the mass spectrometer = 230 °C; temperature of the quadrupole = 150 °C; ionization energy = 70 eV. The identification of components was based on a comparison of their mass spectra and retention time with those of the authentic compounds and by computer matching with the database of the National Institute of Standards and Technology (NIST11) and Adams [70].

### 4.5. In Vitro Antifungal and Anti-Oomycete Activity

The antifungal and anti-oomycete activities were investigated using the agar dilution method [71], incorporating aliquots of stock solutions into the PDA medium to provide final concentrations in the 62.5–1500 µg·mL^−1^ range. Mycelial plugs (ø = 5 mm) were transferred from the margin of seven-day-old fresh PDA cultures in the case of *D. amygdali* and *P. megasperma* and two-week-old fresh PDA cultures in the case of *V. dahliae* to plates filled with the amended media (three plates per treatment and concentration combination; each experiment was carried out twice). Plates containing only PDA medium were used as a control. Radial mycelium growth was determined by calculating the average of two perpendicular colony diameters for each replicate. After incubation in the dark at 25 °C for one week (*D. amygdali* and *P. megasperma*) or two weeks (*V. dahliae*), growth inhibition was calculated according to the formula: ((*d_c_* − *d_t_*)/*d_c_*) × 100, where *d_c_* is the average colony diameter in the control and *d_t_* is the average diameter of the treated colony. The 50% and 90% effective concentrations (EC_50_ and EC_90_, respectively) were estimated using PROBIT analysis in IBM SPSS Statistics v.25 software (IBM; Armonk, NY, USA).

### 4.6. Ex Situ Protection Assays on Artificially-Inoculated Excised Stems

The efficacy of flower extract was tested by artificial inoculation of excised stems in controlled laboratory conditions. Inoculation was performed according to the procedure proposed by Matheron [72], with the modifications described by Sánchez-Hernández et al. [73]. Briefly, young stems of healthy ‘Garnem’ (*P. amygdalus* × *P. persica*) rootstock plants with a 1.5 cm diameter were cut into 10 cm long sections using a grafting knife. The excised stem pieces were immediately wrapped in moistened sterile absorbent paper, and—once in the laboratory—the freshly excised stem segments were first immersed in ethanol 70% for 1 min, then immersed in a NaClO 3% solution for 8 min, and then thoroughly rinsed (five times) with bidistilled sterile water [74]. Some of the stem segments (*n* = 15) were soaked for 15 min in distilled water as a control, while the remaining segments were soaked for 15 min in aqueous solutions to which an appropriate amount of the *S. nigra* flower aqueous ammonia extract was added to obtain MIC (375 μg·mL^−1^), MIC×2.5, and MIC×5 concentrations (*n* = 15 segments/concentration). Alkir^®^ co-adjuvant (1% *v/v*) was added to all solutions (including the control) to facilitate bark penetration of the treatment. The stem pieces were allowed to dry, placed on an PDA Petri dishes, and subsequently inoculated by placing a plug (⌀ = 5 mm), from the margin of 1-week-old PDA cultures of *P. megasperma* on the center of the outer surface of the stem bark. After inoculation, the stem segments were incubated in a humid chamber for 4 days at 24 °C, 95–98% RH. The efficacy of the treatments was assessed by visual inspection for the presence of rotting at the inoculation sites, confirmed under the microscope, both on the outer bark and on the inner bark after careful removal with a scalpel to reveal the cambium. Then, the oomycete strain was re-isolated and morphologically identified from the lesions to fulfill Koch’s postulates.

### 4.7. Statistical Analysis

The results of the in vitro mycelial growth inhibition experiments were statistically analyzed using one-way analysis of variance (ANOVA), followed by a post hoc comparison of means through Tukey’s test at *p* < 0.05, given that the homogeneity and homoscedasticity requirements were satisfied, according to Shapiro–Wilk and Levene tests. R statistical software was used for the statistical analyses [75].

## 5. Conclusions

In this study, the major phytoconstituents of the aqueous ammonia extracts of *S. nigra* aerial parts were investigated by infrared spectroscopy and gas chromatography–mass spectrometry. The most abundant phytocompounds identified in the flower extract were octyl 2-methylpropanoate (10.5%); 3,5-dihydroxy-6-methyl-2,3-dihydropyran-4-one (5.1%); and 2-propyl malonic and dimethylmalonic acids (4.9%); and, in the leaf extract, 1,6-anhydro-β-D-glucopyranose (11.6%); oleic acid (6.3%); and butanoic acid pentyl ester (8%). Concerning the potential of the extracts to act as bioprotectants, in vitro tests against three almond pathogens (*Diaporthe amygdali, Phytophthora megasperma,* and *Verticillium dahliae*) showed higher activity of the flower extract than that of the leaf extract, with effective concentration EC_90_ values in the 241–984 and 354–1322 μg·mL^−1^ ranges, respectively, depending on the pathogen. This effectiveness was higher than those of azoxystrobin, mancozeb, and fosetyl-Al against *D. amygdali* and *P. megasperma* and was second to manzobeb against *V. dahliae*. In view of this promising antimicrobial activity, additional ex situ tests against *P. megasperma* were conducted on artificially infected ‘Garnem’ rootstock excised stems, finding that the flower extract resulted in full protection at a dose of 1875 μg·mL^−1^, ten times lower than those of mancozeb and fosetyl-Al. Collectively, these results encourage further research on the antifungal/oomyceticidal potential of *S. nigra* extracts against phytopathogens.

## Figures and Tables

**Figure 1 ijms-24-01154-f001:**
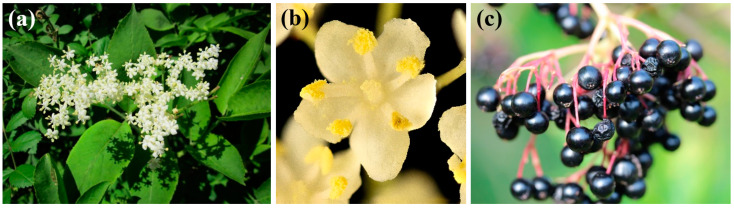
(**a**) Inflorescences and leaves, (**b**) flowers, and (**c**) fruits of *Sambucus nigra*.

**Figure 2 ijms-24-01154-f002:**
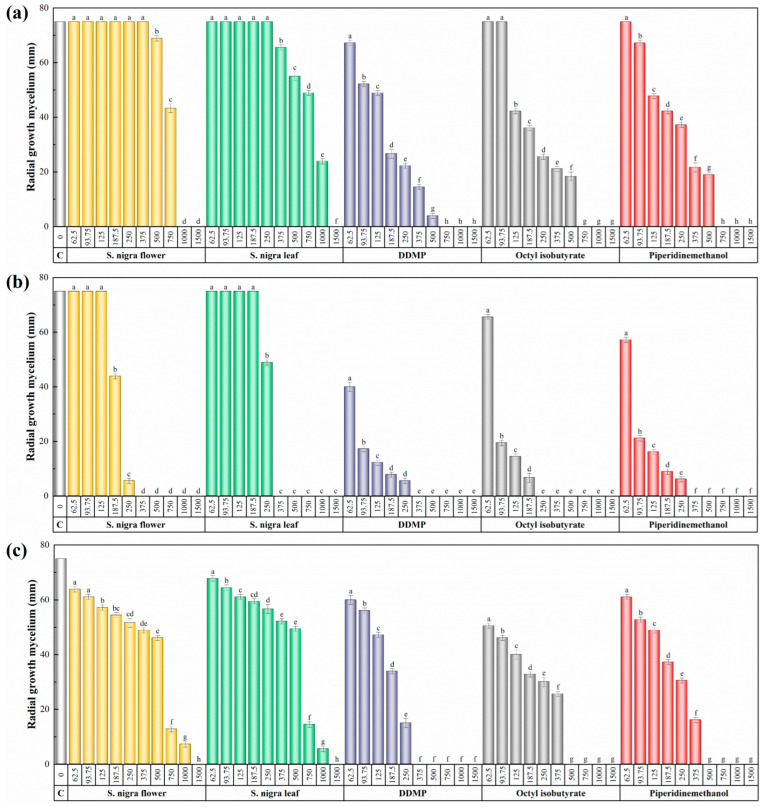
Radial growth of the mycelium of (**a**) *D. amygdali*, (**b**) *P. megasperma*, and (**c**) *V. dahliae* in in vitro assays performed on PDA medium with different concentrations (in the 62.5–1500 μg·mL^−1^ range) of *S. nigra* flower and leaf extracts, as well as some of the main constituents of flower extract. The same letters above concentrations mean that they are not significantly different at *p* < 0.05. Standard deviations are represented by error bars. DDMP = 4H-pyran-4-one, 2,3-dihydro-3,5-dihydroxy-6-methyl-; piperidinemethanol = 1-methyl-2-piperidinemethanol.

**Figure 3 ijms-24-01154-f003:**
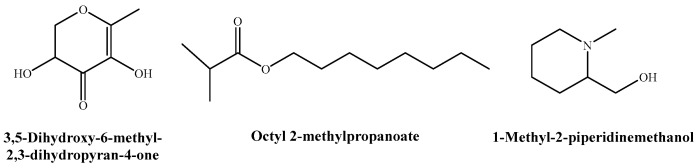
Some of the main phytochemicals identified in *S. nigra* flower aqueous ammonia extract.

**Figure 4 ijms-24-01154-f004:**
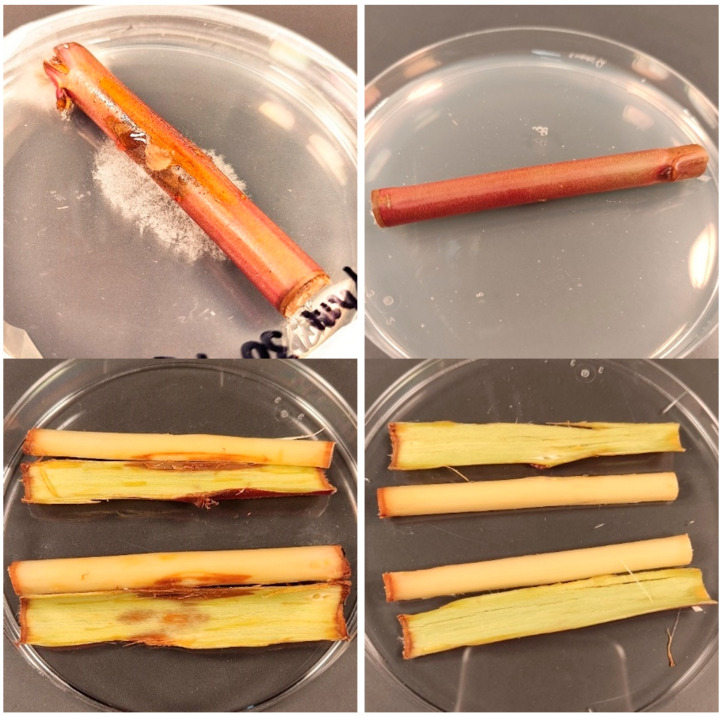
‘Garnem’ stem segments artificially inoculated with *P. megasperma* after 4 days of incubation: (**left**) untreated samples (**top**: outer bark; **bottom**: cambium); (**right**) samples (**top**: outer bark; **bottom:** cambium) treated with *S. nigra* flower aqueous ammonia extract at 1875 μg·mL^−1^ (MIC×5).

**Figure 5 ijms-24-01154-f005:**
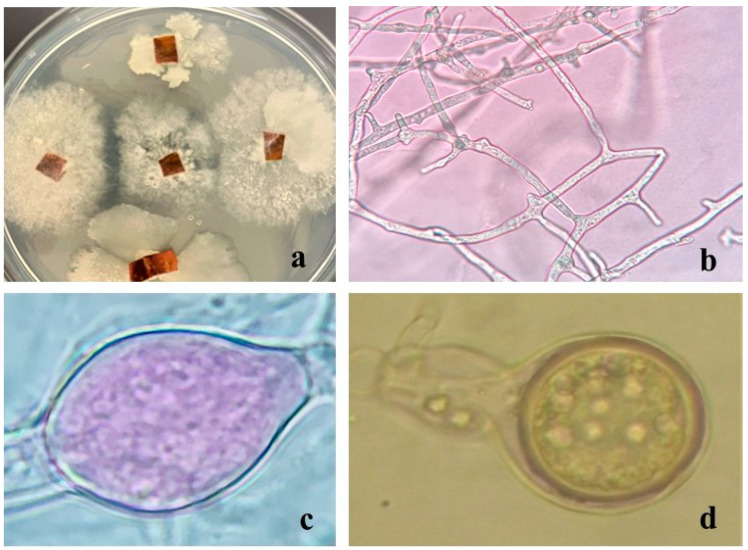
Re-isolation and characterization of the *P. megasperma* strain previously inoculated in ‘Garnem’ stem segments. (**a**) Mycelium in PDA plates subjected to microscopic observations; (**b**) microscopical aspect of somatic hyphae; (**c**) zoosporangia; (**d**) oospore.

**Table 1 ijms-24-01154-t001:** Main bands in the infrared spectra of *S. nigra* flowers and leaves, and those of their aqueous ammonia extracts.

Flower	Flower Extract	Leaf	Leaf Extract	Assignment
	3369			O–H stretching of free hydroxyl groups
3288	3231	3239	3239	bonded O–H stretching (cellulose, hemicellulose, lignin)
2921	2923	2924	2930	–CH_2_ asymmetric stretching of alkyls (cutine, wax, pectin)
2851	2852	2850		–CH_2_ sym. stretch (cutine and wax)/CH_2_–(C6) bend (cellulose)
1733	1733	1733		C=O stretching of alkyl ester
1685	1685	1686		*α,β*-unsaturated carbonyls
1633	1636			skeletal aromatic C=C ring stretching and C=O stretching
1605	1602	1588	1582	aromatic C=C stretching
1519	1516	1522		aromatic skeletal
	1455	1462		CH_2,_ CH_3_ bending/C=C stretching, furan ring (furfural)/O–CH_3_ str.
1388 1376	1387 1375	1389	1389	CH_3_ antisymmetric bending aliphatic C–H stretching in methyl and phenol OH
1312	1315	1314		C–H vibration of the methyl group
1254	1254	1250	1263	C–O–C symmetric stretching/C–O stretching
1143	1164	1163	1154	C–O–C asym. stretching in celluloses; C–O stretching; C–C in-plane
	1100	1121 1100		in plane =C–H bending/C=C stretching/C−O−C stretching in the pyranose ring skeletal (cellulose)
	1047	1045	1046	O–H out plane bending/C–O stretching
1015	1026	1010		C–H bending
919	918	918	914	*β*-glycosidic linkages (glucose units of cellulose chains)
872	873			C–H bending

**Table 2 ijms-24-01154-t002:** Effective concentrations (expressed in μg·mL^−1^) against *D. amygdali*, *P. megasperma*, and *V. dahliae* of *S. nigra* flower and leaf aqueous ammonia extracts, together with those of three of the main constituents of the flower extract.

Treatment	Effective Concentration (µg·mL^−1^)	*D. amygdali*	*P. megasperma*	*V. dahliae*
*S. nigra* flower extract	EC_50_	720.5	193.9	516.1
	EC_90_	981.7	241.2	984.1
*S. nigra* leaf extract	EC_50_	860.3	269.3	539.2
	EC_90_	1322.4	354.6	975.2
DDMP	EC_50_	152.7	63.6	159.9
	EC_90_	482.4	211.1	314.1
Octyl isobutyrate	EC_50_	164.4	87.0	142.4
	EC_90_	647.5	177.2	471.2
Piperidinemethanol	EC_50_	233.4	76.4	185.8
	EC_90_	635.5	225.6	447.0

DDMP = 4H-pyran-4-one, 2,3-dihydro-3,5-dihydroxy-6-methyl-; piperidinemethanol = 1-methyl-2-piperidinemethanol.

**Table 3 ijms-24-01154-t003:** Radial growth of mycelium of *D. amygdali, P. megasperma*, and *V. dahliae* in in vitro assays performed on a PDA medium amended with different concentrations (the recommended dose, 1/10th of the recommended dose, and 10 times the recommended dose) of three commercial synthetic fungicides.

Commercial Fungicide	Pathogen	Radial Growth of Mycelium (mm)			Inhibition (%)		
		Rd/10	Rd *	Rd × 10	Rd/10	Rd *	Rd × 10
Azoxystrobin	*D. amygdali*	33.2	14.2	0	55.7	81.1	100
	*P. megasperma*	48.1	34.3	14.5	35.9	54.3	80.6
	*V. dahliae*	26	24	0	65.3	68	100
Mancozeb	*D. amygdali*	13.4	0	0	82.2	100	100
	*P. megasperma*	61.3	31.3	0	18.3	58.3	100
	*V. dahliae*	0	0	0	100	100	100
Fosetyl-Al	*D. amygdali*	72	9.9	0	4	86.9	100
	*P. megasperma*	75	16.1	0	0	78.6	100
	*V. dahliae*	36	0	0	52	100	100

* Rd stands for recommended dose, i.e., 62.5 mg·mL^−1^ of azoxystrobin (250 g·L^−1^ for Ortiva^®^, azoxystrobin 25%), 1.5 mg·mL^−1^ of mancozeb (2 g·L^−1^ for Vondozeb^®^, mancozeb 75%), and 2 mg·mL^−1^ of fosetyl-Al (2.5 g·L^−1^ for Fosbel^®^, fosetyl-Al 80%). The radial growth of the mycelium for the control (PDA only) was 75 mm. All mycelial growth values (in mm) are average values (*n* = 3).

## Data Availability

The data presented in this study are available on request from the corresponding author. The data are not publicly available due to their relevance to an ongoing Ph.D. thesis.

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
