# Peer review of "Phytochemical Profiling of Sambucus nigra L. Flower and Leaf Extracts and Their Antimicrobial Potential against Almond Tree Pathogens"

_ijms, 2023, doi:10.3390/ijms24021154_

Round 1
Author Response
The paper provides useful information on the phytochemical composition of aqueous ammonia extracts of black elderberry and their potential biocontrol capabilities against three fungal phytopathogens belonging to Ascomycota (Diaporthe amygdali and Verticillium dahliae) and Oomycota (Phytophthora megasperma). The manuscript is well written, with well presented results and adequate discussion. The methodology used is suitable for achieving the set goals. The cited literature is up-to-date and in line with the subject matter. I recommend that the manuscript be accepted after minor revision.
Response: Thank you for your positive feedback.
Q1. L– 345. Since the test phytopathogens used in the study are only of a fungal nature, I would recommend that instead of the term "germicidal activities" the term "fungicidal activities" be used.
Response: We have replaced ‘germicidal’ with ‘fungicidal’, as suggested by the Reviewer.
Q2. Minor corrections to the bibliography are also needed. The following articles are repeated:
L– 625-627 and L– 697-699. Sánchez-Hernández, E.; Martín-Ramos, P.; Martín-Gil, J.; Santiago-Aliste, A.; Hernández-Navarro, S.; Oliveira, R.; González-García, V. Bark extract of Uncaria tomentosa L. for the control of strawberry phytopathogens. Horticulturae 2022, 8, 672 doi:10.3390/horticulturae8080672 (Nos 26 and 56)
L– 638-640 and L– 721-723. Sánchez-Hernández, E.; Buzón-Durán, L.; Cuchí-Oterino, J.A.; Martín-Gil, J.; Lorenzo-Vidal, B.; Martín-Ramos, P. Dwarf pomegranate (Punica granatum var. nana): Source of 5-HMF and bioactive compounds with applications in the protection of woody crops. Plants 2022, 11, 550, doi:10.3390/plants11040550 (Nos 31 and 65)
Response: Thank you for pointing out these mistakes. We have fixed a problem with the ‘traveling library’ in our reference manager, and the indicated references are no longer duplicated.
Reviewer 2 Report
In this article, the authors showed that extracts from the leaves and flowers of Sambucus nigra show antibacterial and antifungal activity against three almond pathogens: Diaporthe amygdali, Phytophthora megasperma and Verticillium dahliae.
The subject is a fascinating and interesting one, as a result of the fact that currently there are very few natural compounds used as antibacterial or antifungal agents. This study is made even more interesting by the fact that Sambucus nigra extracts could be used as antimicrobial prophylaxis to prevent plant infections and thus reduce the use of artificial antibacterial and antifungal agents.
This manuscript is written logically, being well structured and easy to read.
The methods used by the authors are relevant and well selected.
Author Response
In this article, the authors showed that extracts from the leaves and flowers of Sambucus nigra show antibacterial and antifungal activity against three almond pathogens: Diaporthe amygdali, Phytophthora megasperma and Verticillium dahliae.
The subject is a fascinating and interesting one, as a result of the fact that currently there are very few natural compounds used as antibacterial or antifungal agents. This study is made even more interesting by the fact that Sambucus nigra extracts could be used as antimicrobial prophylaxis to prevent plant infections and thus reduce the use of artificial antibacterial and antifungal agents.
This manuscript is written logically, being well structured and easy to read.
The methods used by the authors are relevant and well selected.
Response: Your positive feedback is most appreciated.
Reviewer 3 Report
The authors elaborated a work of extracting some phytoconstituents from the leaves and flowers of S. nigra that were evaluated by FTIR and GC-MS, and subsequent use as antifungal agents and acti-oomycete.
The work is interesting and complete, in addition, the manuscript is well prepared.
I suggest some points to improve the manuscript:
- The title should be shortened.
- FTIR spectra must be added to the manuscript, as well as GC chromatograms.
- In Table S1 and S2 the authors repost several peaks for the same compound, how do they justify this?
- The abstract and conclusions must be improved
Author Response
The authors elaborated a work of extracting some phytoconstituents from the leaves and flowers of S. nigra that were evaluated by FTIR and GC-MS, and subsequent use as antifungal agents and acti-oomycete. The work is interesting and complete, in addition, the manuscript is well prepared.
Response: Your positive feedback is most appreciated.
I suggest some points to improve the manuscript:
Q1. The title should be shortened.
Response: As requested by the Reviewer, the title has been shortened and now reads “Phytochemical Profiling of Sambucus nigra L. Flower and Leaf Extracts and their Antimicrobial Potential against Almond Tree Pathogens”
Q2. FTIR spectra must be added to the manuscript, as well as GC chromatograms.
Response: The FTIR spectra and GC-MS chromatograms have been included in the supplementary material file (Figures S1-S3), as suggested by the Reviewer. The new figures have been mentioned in the main text and the Supplementary Materials section at the end of the manuscript has been updated accordingly.
Q3. In Table S1 and S2 the authors repost several peaks for the same compound, how do they justify this?
Response: GC-MS analyses were outsourced to an accredited laboratory that meets the highest quality standards and whose research personnel only focus on this technique, so equipment/run conditions-related problems would not be expected. Taking into consideration that we conducted no derivatization (which would be one of the possible reasons for the having several peaks for the same compound, given that one can get multiple peaks for the same compound depending on the degree of derivatization, especially in a complex mixture), our interpretation is that -in most cases- we may have small matrix-induced retention shifts (for instance, in Table S1, for hydroquinone; 2-propen-1-amine, N,N-di-2-propenyl-; adenine; n-hexadecanoic acid; and, in Table S2, for glycerin, hydroquinone; propanoic acid, 2-(aminooxy)-; benzoic acid, 3-hydroxy-; butanoic acid, pentyl ester; inositol, 1-deoxy-). However, for some compounds the peaks are further apart in terms of RT and a different explanation should be sought. In the case of oleic acid in Table S2, it is not strange, given that unsaturated compounds provide many examples of the aforementioned behavior. On the other hand, in the case of β-D-glucopyranose, 1,6-anhydro- and 4(1H)-pyrimidinone, 6-hydroxy- (Table S2), we are probably dealing with isomers, either structural isomers or positional isomers, in which the structural changes change the boiling point or other characteristics of the compound sufficiently to change its gas chromatographic retention. We have included a brief explanation as a footnote in Table S1 and Table S2 to bring it to the attention of the reader.
Q4. The abstract and conclusions must be improved
Response: Both the abstract and the conclusions have been rewritten.